# Preliminary study on the application of bioimpedance analysis to measure the psoas major muscle in older adults

Lee-Ping Chu [1,2], Kuen-Tsann Chen[1], Hsueh-Kuan Lu[3], Chung-Liang Lai[4,5], Hsing-Ching Huang[6], Kuen-Chang Hsieh[7,8]*

1 Department of Applied Math, National Chung Hsing University, Taichung, Taiwan, 2 Department of Orthopedics, China Medical University Hospital, Taichung, Taiwan, 3 General Education Center, National Taiwan University of Sport, Taichung, Taiwan, 4 Department of Physical Medicine and Rehabilitation, Puzi Hospital, Ministry of Health and Welfare, Chiayi, Taiwan, 5 Department of Occupational Therapy, Asia University, Taichung, Taiwan, 6 Department of Physical Medicine and Rehabilitation, Taichung Hospital, Ministry of Health and Welfare, Taichung, Taiwan, 7 Department of Research and Development, Starbia Meditek Co., Ltd, Taichung, Taiwan, 8 Big Data Center, National Chung-Hsing University, Taichung, Taiwan

☯ These authors contributed equally to this work.
* abaqus0927@yahoo.com.tw

**Data Availability Statement:** Data are within the Supporting Information files.

**Funding:** This work was supported by grants from the Ministry of Scientific and Technological

## Abstract

For the assessment of sarcopenia or other geriatric frailty syndromes, psoas major area may be one of the primary indicators. Aim to develop and cross-validate the psoas cross-sectional area estimation equation of L3-L4 of the elderly over 60 years old by bioelectrical impedance analysis (BIA). Ninety-two older adults with normal mobility were enrolled (47 females, 45 males), and were randomly divided into a modeling group (MG, n = 62) and validation group (VG, n = 30). Computed tomography (CT) was used to measure the psoas major area at the' L3-L4 lumbar vertebrae height as a predictor. Estimated variables were height (h), whole body impedance ($Z_{whole}$), whole body impedance index ($h^2/Z_{whole}$, WBI), age, gender (female = 0, male = 1), and body weight (weight) by standing BIA. Relevant variables were estimated using stepwise regression analysis. Model performance was confirmed by cross-validation. BIA estimation equation for PMM obtained from the MG was: ($PMM_{BIA}$ = 0.183 $h^2/Z$ – 0.223 age + 4.443 gender + 5.727, $r^2$ = 0.702, n = 62, SEE = 2.432 $cm^2$, p < 0.001). The correlation coefficient r obtained by incorporating the VG data into the PMM equation was 0.846, and the LOA ranged from -4.55 to 4.75 $cm^2$. $PMM_{BIA}$ and $PMM_{CT}$ both correlate highly with MG or VG with small LOA. The fast and convenient standing BIA for measuring PMM may be a promising method that is worth developing.

## Introduction

Loss of skeletal muscle mass and its function is one of the main reasons why the elderly suffer from many acute and chronic diseases. These diseases include endocrine, metabolic disorders, infectious diseases, immune diseases, vascular diseases, hematological diseases and malignant tumors [1]. Therefore, an accurate and convenient quantitative measurement method of skeletal muscle is needed to diagnose and study the changes of skeletal muscle mass in such patients [2–4].

Development, Higher Ecation and Information Society with the award number of PG11001-0566. The funders had no role in study design, data collection and analysis, decision to publish, or preparation of the manuscript.

**Competing interests:** One of the authors (KCH) is employed by Starbia Meditek Co., Ltd. This does not alter our adherence to all the PLoS ONE policies on sharing data and materials. There are no patents, products in development nor marketed products to be declared. The other authors declare no conflict of interest.

Trunk muscles provide a key support mechanism in maintaining body stability and balance during limb movement [5]. A strong and stable trunk can provide a solid foundation for torque-generating limb movements [6, 7]. Among muscles located in the trunk, the paired psoas major (combination of iliacus and psoas muscles) in the core is the only muscle that connects the lumbar spine to the bilateral lower extremities [8]. Therefore, the psoas major of the core muscle group is clearly the main muscle playing an important role in the motor function of the trunk and extremities [9].

Core muscle cross-sectional area (CSA) can be used as an important indicator of health status in morphometric analysis [10, 11]. Aerobic exercise and resistance training are the foremost ways to increase the CSA of muscle fibers [12]. The psoas is a large abdominal muscle that forms part of the core and assists in lateral rotation and abduction of the hip joint [13]. The CSA of the psoas major has been used in several studies to predict whole-body muscle mass [14], cardiorespiratory fitness [10], sarcopenia [15], and the prognosis of surgery in older adult patients [16, 17].

Muscles can be divided into surface muscles and deep muscles. The psoas major is a deep muscle. Deep muscles cannot be measured by contact, and only electronic signal processing and imaging techniques can be used to diagnose or quantify deep muscles. Numerous studies have provided substantial data on the size and anatomical variability of the psoas major [9]. The use of ultrasound to measure the thickness of the psoas major muscle is a convenient method [18]. However, imaging methods such as computed tomography and magnetic resonance imaging can be reliably used to quantify the size of the psoas major muscle [19, 20]. However, between the above two methods, a slight difference is still noted in the measurement results of the normal and diseased (calcifications and air bubbles in abscesses) psoas major muscle [21]. Marras *et al.* [22] used anthropometry to estimate the cross-sectional area of the lumbar core and psoas major muscle.

For the past three decades, BIA has been widely used for the estimation of body composition in various fields, including sports science, clinical research, health care and other fields, due to its advantages of convenience, non-invasiveness, and rapid results [23]. Items for estimating and measuring body composition include body composition components such as the whole body, individual limb segments, trunk body fat mass, muscle mass, body fluid, and intracellular and extracellular fluids [24]. However, studies on the measurement of core muscle groups or various muscle groups in the waist remain limited.

In view of the above problems, this study intended to use the CSA of the psoas major muscle of the core muscle group as the estimation target. Variables such as anthropometric measurements and BIA measurements were used as predictors. The study purpose was to establish and validate an estimation model for the area of the psoas major in older adults.

## Materials and methods

### Participants

The study was conducted in accordance with the Declaration of Helsinki, and approved by the Ethics Committee of Tsao-Tun Psychiatric Center, Ministry of Health and Welfare (protocol code IRB-10943 and date of approval 22-12-2020). A total of 92 older adults (aged 60 years and older) in central Taiwan were recruited for this study through their response to posters placed in communities within the northern Taiwan area. The subjects were tested using the non-random purposive sampling method. Potential participants were healthy adults who came to the hospital for their free NHS health check-up. All included participants provided signed informed consent as required by the Human Experiment Ethics Committee. Trials registered before the onset of patient enrollment at the Chinese Clinical Trial Registry

(ChiCTR2100051573). The study was conducted at a community hospital, from September 2021 to December 2021. Generally healthy adults aged 60 and older with normal mobility were included; those with one or more chronic diseases (e.g., cancer, renal failure, hepatitis, chronic lung disease, etc.) were excluded.

## Impedance measurement

Participants were not permitted to drink alcoholic beverages 48 hours before the test, did not use diuretics for seven days before the test, and did not participate in intense exercise training 24 hours before the test. Impedance measurement required subjects to use a standing octopolar (i.e., eight electrical charges) plate bioimpedance analyzer BC418MA (Tanita Co., Tokyo, Japan). During the test, the participant stands on the base platform, with the soles of the feet touching the sensing and transmitting electrode pairs naturally under the body weight pressure. The handles embedded with the transmitting and sensing electrode plates were held with both hands. **In Fig 1(A),** the BC418MA analyzer applies a single frequency of 50 KHz and measurement current of 0.55 mA to measure the impedance of the right upper limb, right lower limb and the whole body, respectively (represented as $Z_{rightarm}$, $Z_{rightleg}$, $Z_{whole}$). Measures of the whole body and the right upper and lower extremities were subtracted to represent impedance of the whole trunk (denoted as $Z_{trunk}$). The square of the height (unit: $cm^2$) was divided by impedance of the whole body and the trunk, which was equal to the bioimpedance index of the whole body and the trunk (represented as WBI, TBI).

The coefficient of variation of impedance measurements of the right hand-to-foot current flow path was assessed using within-day and between-days measurements. Five male and five female participants repeated the impedance measurement 10 times within one hour of the day. Between-days impedance measurements were performed within the same time period over five days.

## Anthropometry

The body weight (kg) of each participant was measured by the same experienced research assistant. The body weight of the participants was measured to the nearest 0.1 kg using a body composition analyzer BC418MA. The participant's barefoot height was measured to the

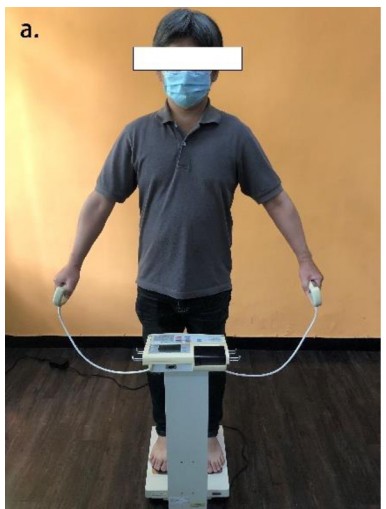
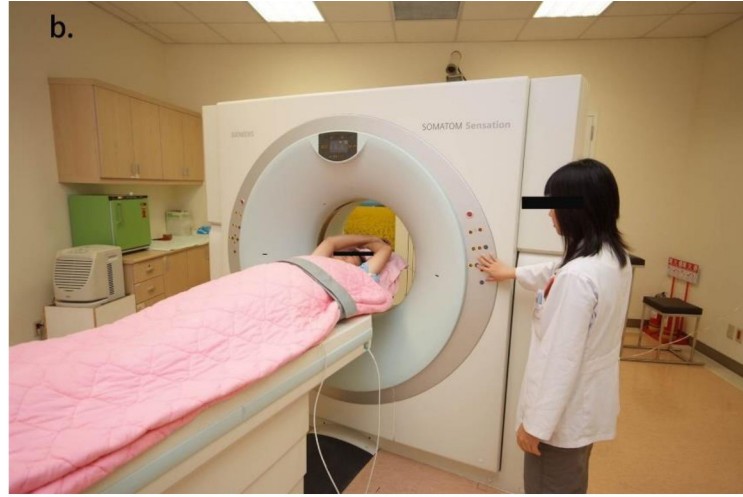

**Fig 1. Impedance measurement and computed tomography.** (a) Whole body impedance measurement with standing bioimpedance analyzer (Taitan BC418MA); (b) Cross-sectional scan of abdomen with Somatom Sensation 64 CT system.

nearest 0.5 cm using a height ruler (Holtain Ltd., Crosswell, Wales, UK). Body mass index (BMI) was defined classically as weight (kg) divided by height (meters) squared. Waist circumference (WC) was measured at the level of the navel. Hip circumference (HC) was measured at the hip and the largest circumference around the buttocks [25]. Both WC and HC were measured using a standard tape measure to the nearest 0.1 cm. Each anthropometric measurement was performed by trained observers with all participants wearing light-weight cotton/polyester blend hospital gowns and minimal underwear. Measuring tools were calibrated weekly.

## Computed tomography

A 64-slice computed tomography scanner (Somatom Sensation 64 CT system, Siemens Corp., Germany) was used for abdominal CT scans. Data were analyzed using Syngo CT2005A software. Each participant lay in the center of the CT scanner while the abdominal segment of 2–5 lumbar vertebrae was measured. The main scanning parameters of CT measurement were set as X-ray tube voltage: 120Kv, a tube current of 120 mAs, X-ray beam width of 1.5 mm, scanning time was 0.5 s, slice thickness was 5 mm, slice increment was 2mm, image reconstruction kernel index was B20. In **Fig 1(B)**.

Two image analysts (observers) were trained by radiologists in human anatomy associated with the region of the L3-L4 lumbar spine. A fixed image analysis program was used to calculate fat, and CSA of the psoas major at L3-L4. Image processing software Slice-O-Matic version 4.3 (Tomovision, Magog, QC, Canada) was used, and the image analysis program Slice-O-Matic [19] was used for quantitative analysis of the image area. DICOM (Digital Imaging and Communications in Medicine) was the image format used. Image analysts applied Slice-O-Matic to the image opening and circular section of lean tissue, and the area within the delineated boundary was calculated by the software (as shown in **Fig 2**). The rectus abdominis (RA) can be clearly observed in the same axial slice. The cross-sectional area of the PMM in this study is the sum of the left and right PMMs, as shown in **Fig 2**. The PMM, abdominal CSA (PMM$_{CT}$), abdominal visceral fat area, and abdominal subcutaneous fat area were manually segmented at L3-L4 waist level.

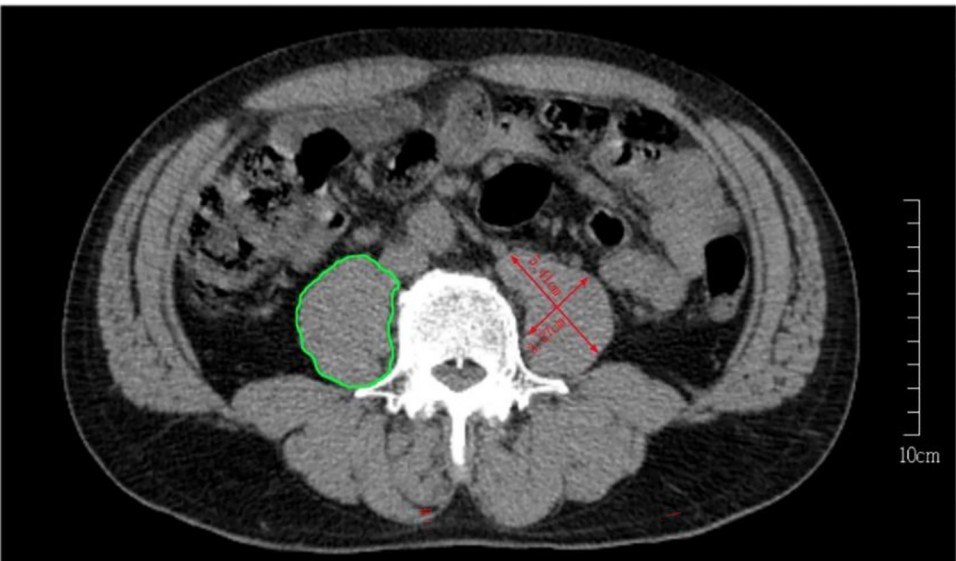

**Fig 2. Computed tomography image of spine L3-L4.** The red line is the length calculated by the software, and the area surrounded by the green line is the left psoas muscle using the software Slice-O-Matic.

## Reproducibility

To test the reproducibility of manual analysis of PMM, abdominal visceral fat, abdominal subcutaneous fat, and CSA were measured by the same observer. Twenty randomly selected axial slices were performed at three different times to measure the different parts of the CSA. The mean ± standard deviation (mean ± SD) of PMM was 15.2 ± 3.8 cm$^2$, and the coefficient of variation (CoV) was 0.43%.

## Statistical analysis

Descriptive data are expressed as mean plus standard deviation (mean ± SD), and are shown with ranges. Data were tested for normal distribution using the Kolmogorov-Smirnov test. All statistical analyses were performed using SPSS V20 (Statistical Package for the Social Sciences, IBM SPSS statistics for Windows; IBM Corp., Armonk, NY, USA). Power analysis uses the G-Power analysis software program (G Power 3.1.9, University of Dusseldorf, Germany) to determine the minimum number of samples required to establish the most regression equation (calculated with alpha = 0.05, two-tailed, power = 0.95, Number of predictors = 4), the minimum sample size is 60. The significance level was set at $p < 0.05$ (two-tailed). Spearman correlation coefficient analysis was used to describe the correlation between variables. Respectively, two-thirds and one-third of the total subjects were randomly divided into a MG and a VG. Height, weight, age, gender, WC, HC, BMI, trunk bioimpedance index (TBI), and whole body bioimpedance index (WBI) were used as predictor variables. CT-derived psoas major area (PMM$_{CT}$) was the response variable. In applying stepwise regression analysis in the MG, the parameters—forward (F$_{in}$ = 4.00), backward (F$_{out}$ = 3.99) were used to obtain the estimated variables selected by the PMM estimation equation (PMM$_{BIA}$), and to obtain the corresponding regression coefficients, the standard estimate error (SEE), the pure errors (square root of mean of squares of differences between observed and predicted values) and the coefficient of determination (r$^2$) were used to evaluate the performance of the estimation equation. When the correlation between estimators was too high, the variance inflation factor (VIF) $\geq 4$ was applied to remove the estimator from the estimator equation. Using VG data, the obtained PMM$_{BIA}$ measurement results and PMM$_{CT}$ were analyzed by correlation and Bland-Altman plots, respectively. The ordinary least products regression analysis was used to examine the relationship between the PMM$_{CT}$ and PMM$_{BIA}$ and to determine the fixed bias proportional bias [26].

## Results

The subjects were randomly divided into an MG of 62 participants and a VG of 30 participants. The MG included 30 males (age: 75.5 ± 8.4 years old, BMI: 25.3 ± 3.1 kg/m$^2$), and 32 females (age: 70.3 ± 7.0 years old, BMI: 24.9 ± 3.8 kg/m$^2$). The VG included 15 males (age 74.6 ± 8.5 years old, BMI: 23.8 ± 3.3 kg/m$^2$) and 15 females (age: 70.1 ± 6.6 years, BMI: 23.4 ± 4.2 kg/m$^2$). The PMM$_{CT}$ was 19.1 ± 4.7 cm$^2$ in males and 12.6 ± 2.6 cm$^2$ in females, representing normal distribution. The measurement results of PMM$_{CT}$ and other human parameters are shown in **Table 1.** The within-day CoV for whole-body impedance evaluated in six subjects was 1.0% - 1.9%, the corresponding between-days CoV was 1.4% - 2.1%.

**Table 1** shows participants' baseline data. The impedance value of each limb segment and the CSA of visceral fat, subcutaneous fat and PMM at the lumbar vertebrae L3-L4 showed significant differences between the PMM area of males and females (19.1 ± 4.7 cm$^2$ and 12.6 ± 2.6 cm$^2$, respectively). The PMM area of males was significantly higher than that of females ($p < 0.001$).

**Table 1. General physical characteristics and body composition data for male and female subject.**

| Items | Female (n = 47) | | Male (n = 45) | | |
| --- | --- | --- | --- | --- | --- |
| | mean±SD | Range | mean±SD | Range | P |
| Age (years) | 70.2±6.8 | 60.4–87.6 | 75.2±8.4 | 60.5–88.5 | >0.05 |
| Height (cm) | 155.5±6.4 | 143.0–167.0 | 166.1±6.6 | 151.5–183.0 | < 0.01 |
| Weight (kg) | 57.7±9.1 | 42.0–77.0 | 68.6±11.0 | 47.0–90.0 | < 0.001 |
| BMI (kg/m$^2$) | 23.8±3.8 | 17.6–34.2 | 24.8±3.2 | 18.8–33.9 | < 0.01 |
| Waist (cm) | 84.5±9.5 | 67.0–103.2 | 91.3±7.1 | 78.0–101.5 | <0.05 |
| Hip (cm) | 97.5±6.8 | 84.5–108.5 | 97.9±5.3 | 87.0–107.0 | >0.05 |
| WHR | 0.87±0.06 | 0.77–1.00 | 0.93±0.04 | 0.88–1.01 | <0.05 |
| **Bioimpedance** | | | | | |
| $Z_{Whole}$ (ohm) | 661.0±71.9 | 469.7–838.0 | 542.2±49.9 | 449.0–673.7 | < 0.01 |
| $Z_{rightarm}$ (ohm) | 371.9±48.2 | 269.3–491.0 | 295.0±32.6 | 233.0–390.3 | < 0.01 |
| $Z_{rightleg}$ (ohm) | 247.8±29.9 | 172.0–304.7 | 211.9±21.7 | 176.7–266.3 | < 0.01 |
| $Z_{trunk}$ (ohm) | 41.3±26.1 | 5.0–134.0 | 35.2±19.1 | 4.5–97.3 | < 0.01 |
| **CT** | | | | | |
| $SAT_{CT}$ (cm$^2$) | 174.5±74.2 | 73.5–347.8 | 118.4±45.7 | 36.9–215.2 | < 0.01 |
| $VAT_{CT}$ (cm$^2$) | 89.8±53.4 | 20.0–251.5 | 143.2±66.1 | 19.6–293.0 | < 0.01 |
| $PMM_{CT}$ (cm$^2$) | 12.3±3.2 | 5.0–19.3 | 18.9±4.6 | 8.9–27.9 | < 0.01 |
| $ASCA_{CT}$ (cm$^2$) | 512.6±124.6 | 313.8–807.6 | 565.9±112.0 | 348.2–758.7 | < 0.01 |
| **Modeling group** | Female (n = 32) | | Male (n = 30) | | |
| Age (years) | 70.3±7.0 | 60.4–87.6. | 75.5±8.4 | 61.1–73.0 | >0.05 |
| Height (cm) | 155.1±6.3 | 143.0–167.0 | 166.1±6.3 | 151.5–176.0 | < 0.01 |
| Weight (kg) | 57.8±8.8 | 42.0–75.0 | 69.9±10.7 | 47.0–90.0 | < 0.001 |
| BMI (kg/m$^2$) | 24.0±3.6 | 17.6–33.3 | 25.3±3.1 | 18.8–31.9 | < 0.01 |
| $BI_{Whole}$ (cm$^2$/ohm) | 36.9±4.5 | 27.2–55.6 | 51.6±6.4 | 38.7–62.0 | < 0.001 |
| $BI_{trunk}$ (cm$^2$/ohm) | 973.3±1569 | 191–6683 | 970±841 | 269.6–4930 | < 0.001 |
| $Z_{trunk}$ (ohm) | 40.2±24.3 | 5.0–124.0 | 37.4±21.4 | 4.5–97.3 | < 0.001 |
| $Z_{Whole}$ (ohm) | 659.8±76.5 | 469.7–838.0 | 540.3±54.1 | 449.0–673.7 | < 0.001 |
| $PPM_{CT}$ (cm$^2$) | 12.6±2.6 | 6.7–18.5 | 19.1±4.7 | 8.9–27.9 | < 0.001 |
| **Validation group** | Female (n = 15) | | Male (n = 15) | | |
| Age (years) | 70.1±6.6 | 60.8–84.8 | 74.6±8.5 | 60.5–88.5 | >0.05 |
| Height (cm) | 156.5±4.4 | 150.0–166.5 | 166.3±7.5 | 155.0–183.0 | < 0.01 |
| Weight (kg) | 57.4±10.2 | 42.0–77.0 | 65.7±11.3 | 53.0–85.0 | < 0.001 |
| BMI (kg/m$^2$) | 23.4±4.2 | 17.9–34.2 | 23.8±3.3 | 19.5–28.7 | < 0.01 |
| WBI (cm$^2$/ohm) | 37.3±4.3 | 31.5–44.4 | 51.07±6.5 | 40.5–65.9 | < 0.001 |
| TBI (cm$^2$/ohm) | 943.3±1 | 342–5633 | 953±734 | 289–4330 | < 0.001 |
| $Z_{trunk}$ (ohm) | 42.3±27.4 | 9.0–134.0 | 35.5±19.8 | 7.5–95.3 | < 0.001 |
| $Z_{Whole}$ (ohm) | 663.7±63.4 | 515.0–741.0 | 546.0±41.9 | 493.0–621.0 | < 0.001 |
| $PPM_{CT}$ (cm$^2$) | 11.5±4.1 | 5.0–19.3 | 18.6±4.5 | 12.4–27.6 | < 0.001 |

Data presented as mean ± SD, WHR: waist to hip ration, BMI: body mass index, SAT: subcutaneous fat, VAT: visceral fat area, PMM: Psoas Major muscle, CT: computer tomography (L3-L4); SD: standard deviation, WBI: whole body bioimpedance index, TBI: trunk bioimpedance index

The applied measurement variables included height, age, gender, WBI TBI, WC, HC, and BMI as independent variables for stepwise regression analysis. After calculation, height, WC, HC, BMI, and TBI were excluded; WBI, age, and gender were included. The PMM estimation

equation is as follows.:

$$PMM = 5.727 + 0.183\,h^2/Z - 0.223\,age + 4.443\,gender$$

$$(r^2 = 0.709, SEE = 2.724\,cm^2, n = 61, p < 0.001) \tag{1}$$

**Fig 3** represents the regression line obtained by the Eq (1), including the distribution and its mean difference, the limit of agreement (LOA) in the distribution diagram and the Bland-Altman plots in the MG. The regression equation in **Fig 3(A)** is $PMM_{CT} = 0.709\,SFA_{BIA}$

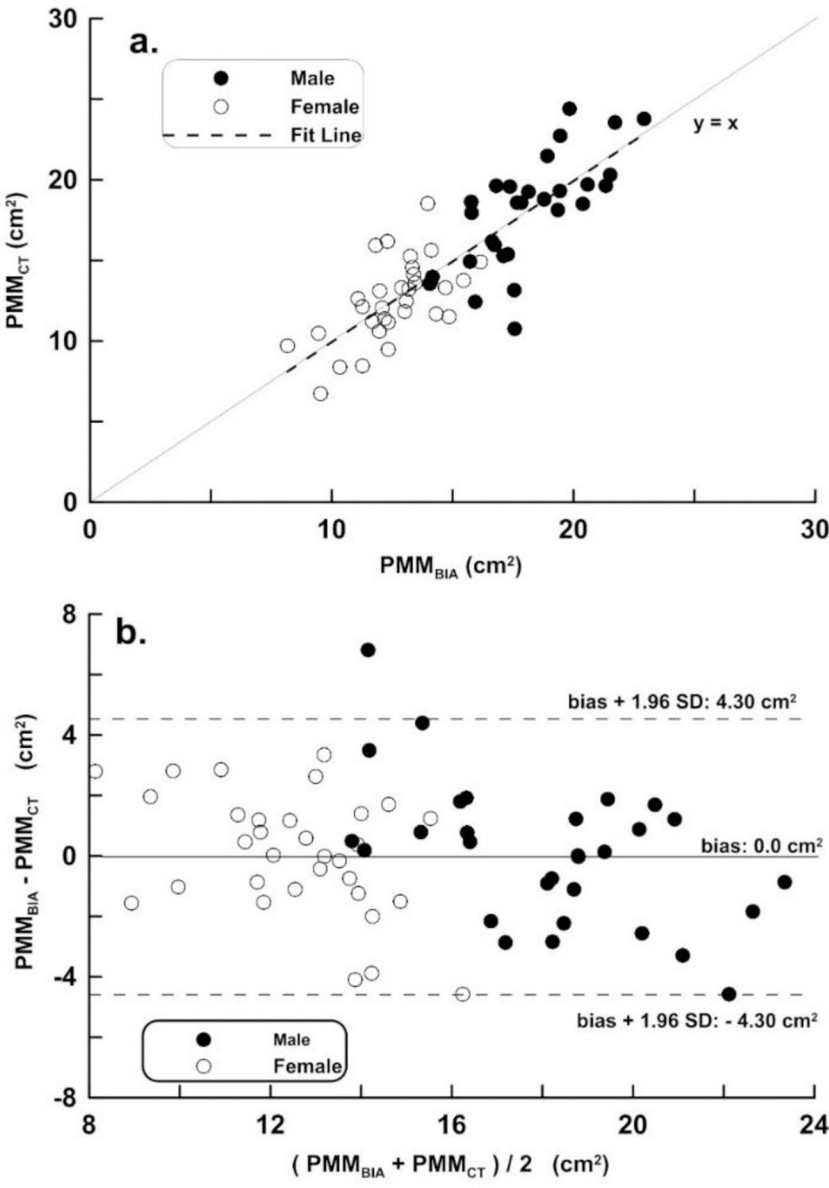

**Fig 3. Scatter and Bland-Altman diagram of modeling group.** Correlation analysis (top) and difference analysis (bottom) between psoas major muscle (PMM) by computed tomography ($PMM_{CT}$) and bioimpedance analysis (BIA) in the modeling group (MG). The difference ($PMM_{CT}-PMM_{BIA}$ of Bland-Altman) is the mean of the corresponding measurements of PMM to CT and BIA. Black filled circles represent males and black open circles represent females.

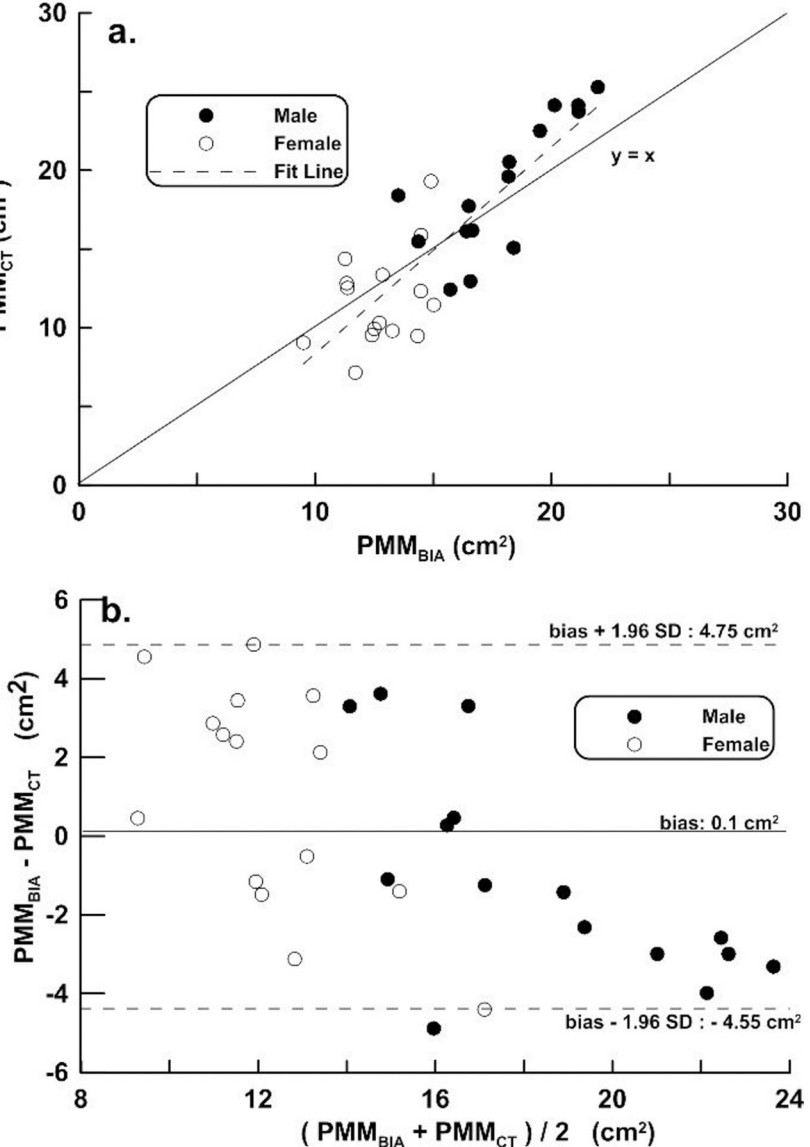

**Fig 4. Scatter and Bland-Altman diagram of validation group.** Correlation analysis (top) and difference analysis (bottom) between psoas major muscle (PMM) by computed tomography ($PMM_{CT}$) and bioimpedance analysis (BIA) in the Validation group (VG). The difference ($PMM_{CT}$–$PMM_{BIA}$ of Bland-Altman) is the mean of the corresponding measurements of PMM to CT and BIA. Black filled circles represent males and black open circles represent females.

+ 4.493, r = 0.708, SEE = 2.34 cm$^2$, p < 0.001. In **Fig 3(B)**, bias ± 1.96 SD was -4.30 to 4.30 cm$^2$.

In **Fig 4**, the regression line obtained by applying Eq (1) includes distribution and its mean difference, LOA on the distribution plot and Bland-Altman plots on the VG. The regression line equation in **Fig 4(A)** is $PMM_{CT}$ = 0.689 $PMM_{BIA}$ + 5.033, r = 0.846, and SEE = 2.45 cm$^2$. In **Fig 4(B)**, bias ± 1.96 SD was -4.55 and 4.55 cm$^2$, respectively.

The correlation coefficients between the response variable $PMM_{CT}$ and each estimated variable were WBI (r = 0.77), gender (r = 0.61), height (r = 0.67), weight (r = 0.63), WC (r = 0.49), BMI (r = 0.44), HC (r = 0.30), TBI (r = 0.03), and age (r = -0.47), respectively.

## Discussion

In the present study, the WBI correlated highly with the CSA of the psoas major at L3-L4 lumbar spine height. This key finding supports our hypothesis that the WBI from bioimpedance measurements can be used as an estimated variable for estimating the PMM of older adults at L3-L4 lumbar spine height.

Several imaging methods use a single scan from the 4th to 5th lumbar vertebrae (L4-L5) to determine the distribution of lumbar muscles and measurement of the psoas major [27]. In the present study, the L3-L4 position was also used for measurement. However, whether these two positions are the best positions to represent the total volume of the psoas major remains to be further explored.

Englesbe *et al.* [11] applied CT to measure the total CSA of the PMM at the fourth lumbar vertebra in 1453 subjects. The results showed that the PMM of male and female subjects was inversely proportional to age, and the correlation coefficients were male, r = 0.485; female, r = 0.481, which were similar to the findings of the present study.

Using ultrasound to measure the CSA of the psoas major and its derived indexes can be applied to evaluate the physiological status and disease severity of sarcopenia patients [28]. Anthropometric measurements can also be applied to estimate the CSA of the psoas major [22]. The above two methods are relatively simple and easy to apply, but measurement accuracy is not high, which limits their application. A certain amount of training is needed to use ultrasound to measure the CSA of the psoas major before the measurement results can be of practical value.

Ishiguro *et al.* [29] used BIA to measure the trunk bioimpedance index (TBI) to estimate the trunk skeletal muscle volume in 28 male participants. Results of that study showed a highly significant correlation between TBI and trunk skeletal muscle mass and volume (r = 0.884, p < 0.05). Because the impedance of the trunk is difficult to measure and standard measures are not available, differences may be shown in estimation results between studies even though the current BIA method is used to estimate the muscles or lean muscle of the trunk [30, 31]. The association between TBI and $PMM_{CT}$ in the present study was limited (r = 0.03) but the WBI using whole-body measurements correlated with $PMM_{CT}$ (r = 0.77). Multivariate linear regression analysis also revealed a significant correlation between the WBI and PMMCT—representing a key finding in the present study and demonstrating the value of BIA in estimating the area of the psoas major in older adults.

This study was a preliminary study conducted to explore the model establishment and feasibility of BIA in PMM measurement. To evaluate the ability of BIA to measure body composition, the pilot study usually starts from a single frequency. The simplicity of using a single frequency makes it easy to focus on bioimpedance and to explore the relationship between the response variables. Compared with single-frequency BIA measurement, multi-frequency BIA can be shown to provide more reliable results than single-frequency BIA for soft tissue lean mass [32]. However, more research in needed in the future to verify, while multi-frequency BIA or BIS (bioelectrical impedance spectroscopy) is worth discussing relative to PMM measurement. The $PMM_{BIA}$ equation constructed in MG or the correlation coefficient brought into VG compared with $PMM_{CT}$ all correlated highly. This is an important reference when establishing new measurement models or equations. In addition, for verification, it is still necessary to incorporate the Bland-Altman plot [33] and other statistical indicators to show agreement between the different methods. The estimation of BIA in PMM was verified by MG in this study, and the correlation and consistency have good results. But this study is Preliminary research on the BIA estimation of PMM. In addition to the consideration of increasing the number of subjects, it is also necessary to confirm the repeatability of this research through research in other laboratories, so as to increase the application value of BIA in PMM.

**Table 2. Multiple regression analysis results for PMM$_{CT}$, based on impedance index and anthropometric.**

| PMM$_{CT}$, Total subject (n = 62) | | | | | | | |
|---|---|---|---|---|---|---|---|
| h$^2$/Z | +Age | +Gender | Intercept | SEE(cm$^2$) | r$^2$ | VIF | B |
| .370±.047* | - | - | -.671±2.101 | 3.225 | .601 | 3.187 | .352 |
| .386±.044** | -.164±.048** | - | 10.568±3.814* | 2.729 | .632 | 1.160 | -.389 |
| .183±.071** | -.223±.047** | 4.443±1.287** | 5.727±1.488** | 2.432 | .702 | 3.443 | .491 |

PMM$_{CT}$, CT-measured Psoas Major Section Area; Regression coefficient estimate ± SEE; r$^2$, coefficient of determination.

p* < 0.01; p** < 0.001; β, standardized coefficient; VIF: variance inflation factor; SEE, standard error of the estimate.

Daytime impedance changes in whole-body or hand-to-foot modes are small relative to changes in arm and leg segment impedance measurements. If using segment impedance measurements to estimate body fat percentage, body water content, or the PMM of body composition, it is especially necessary to measure impedance values at regularly scheduled times [34].

The present equation is a surrogate method using BIA to indirectly estimate PMM for future clinical purposes (e.g., sarcopenia). A close relationship is noted between the BIA method and the ratio of Drills and Contini [35] as well as Dempster body segment mass and support/inertia parameters [36]. Therefore, the logic and results of the present study using BIA to measure a single muscle in normal subjects are reasonable. Relative to diagnostic accuracy, applicability of the BIA method needs to be further confirmed. In particular, considering the individual variability in measurements between PMM$_{CT}$ (reference/gold standard) and PMM$_{BIA}$ was as high as 4.5 cm$^2$ in the present study. Considering the existing use of PMM sarcopenia or debilitating values are the thresholds for evaluation [37, 38]. Furthermore, the study did not include sarcopenic patients. In terms of accuracy and sensitivity, the establishment of measurement methods and models proposed in this preliminary study cannot be used to replace CT or MRI detection in clinical practice at this development stage. The PMM estimation variables and their corresponding regression coefficients in **Table 2.** can be used to express the sensitivity of the PMM estimation model. This study still has limitations on the measurement accuracy of PMM, but it is expected that adding other measurement parameters to the existing single-frequency BIA estimation equation may increase its measurement accuracy. In the future, it can be considered to add measurement parameters such as body circumference measurement in 3D human body scanning [39, 40], local impedance measurement of the waist or back [41], or multi-frequency impedance or bio-impedance spectrum analysis [32] to optimize the measurement model of PMM. Opportunities will likely arise in the future to meet the requirements of clinical application. In order to increase the accuracy of the measurement of impedance of PMM, the multiple electrodes of segmental BIS (or BIA) must be carefully aligned. We findings are only applicable for this single-frequency BIA equipment and that further studies are required to test its applicability to other models/equipment or if a 50 kHz frequency obtained from multi-frequency or BIS equipment were used.

Compared with hospitalized elderly injured patients, the cross-sectional area of psoas major muscle increases by 1 cm$^2$, and the dependent living decreases by 20% [42], which is of great importance in its application. Psoas major muscle CSA in L3-L4 Level relative to other muscles, such as psoas major muscle in L4-L5, L5-S1, Multifidus in L3-L4, L4-L5 and L5-S1, and the correlation of skeletal muscle are 0.877, 0.646, 0.500, 0.585, 0.596, and 0.723, respectively [43]. Based on studies of quantitative indicators of generalized muscle atrophy in patient populations, psoas measurement has not yet to pass the test of all technical and clinical value. Of the numerous studies on psoas sarcopenia, it has been established that the need for standardized thresholds are important representations for predicting morbidity or mortality, but

currently these thresholds are reported in different units of measurement. Therefore, general conclusions cannot be drawn [44]. According to the estimation equation of psoas major muscle in this study, the SEE% of women and men are 19.3% and 12.7%, respectively. This result still has a certain space for practical clinical application.

It has been confirmed that WBI can be used to estimate the lean mass of the whole body [45]. The lean mass of the whole body has a certain correlation with the psoas major [43]. Generally speaking, the muscle mass of the lower limbs will significantly decrease with increasing age. Ikezoe et al. [46] measured the thickness of 10 muscles of the lower extremity, showing minimal atrophy in the soleus muscle. Soleus muscle thickness did not differ significantly between young and old. Among the lower extremity skeletal muscles, the psoas has the highest rate of loss with age. Using WBI, are the variables of thigh and trunk muscles related to the estimation of psoas major muscle mass? In the future, data obtained from tools such as DXA, CT, and multi-limb segment BIA can be used to conduct correlation research.

Results of the present study may be limited in terms of generalizability, including mainly that the participants were all Taiwanese older adults from the community with normal mobility and daily activities, suggesting that the estimation equations derived from this study may only be applicable to Asians aged 60 years or older, and relatively healthy or without chronic disease. If it is necessary to estimate the CSA of the psoas major in certain patients, other ethnicity or subjects of different ages, the applicability of BIA equation may only be confirmed after subsequent rigorous verification. The application of BIA has a great advantage in that muscle mass can be easily measured in a clinical setting. Although there are advantages in estimating muscle mass of the whole body and each segment. At this stage, this approach cannot exactly distinguish individual muscles.

## Conclusions

In the present study, exploration of applying BIA to the measurement of PMM in older adults in Taiwan revealed that BIA can readily be used for preliminary measurement of the estimation of CSA of the psoas major in older adults. Whole-body impedance index (WBI) can serve as an important estimator of the psoas major, and with advantages such as accuracy of estimation and convenience of use, BIA provides a great opportunity for routine use in the measurement and application of PMM for future clinical purposes.

**Informed Consent Statement:** All participants were able to perform the experiment only after they had understood the experimental procedures and related rights and interests explained by the research assistants. All included participants provided signed informed consent as required by the Human Experiment Ethics Committee.

## Supporting information

**S1 Data.**
(XLSX)

## Author Contributions

**Conceptualization:** Lee-Ping Chu, Kuen-Chang Hsieh.

**Data curation:** Lee-Ping Chu.

**Formal analysis:** Chung-Liang Lai.

**Funding acquisition:** Chung-Liang Lai.

**Investigation:** Hsueh-Kuan Lu.

**Methodology:** Lee-Ping Chu, Hsing-Ching Huang.

**Project administration:** Hsueh-Kuan Lu.

**Software:** Kuen-Tsann Chen.

**Supervision:** Kuen-Chang Hsieh.

**Validation:** Kuen-Tsann Chen, Chung-Liang Lai, Hsing-Ching Huang.

**Visualization:** Kuen-Tsann Chen.

**Writing – original draft:** Lee-Ping Chu, Kuen-Tsann Chen.

**Writing – review & editing:** Kuen-Chang Hsieh.

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
