## [Decision Letter · Decision Letter 0]

2 Nov 2022

PONE-D-22-26490Preliminary study on the application of bioimpedance analysis to measure the psoas major muscle in older adultsPLOS ONE

Dear Dr. Hsieh,

Thank you for submitting your manuscript to PLOS ONE. After careful consideration, we feel that it has merit but does not fully meet PLOS ONE’s publication criteria as it currently stands. Therefore, we invite you to submit a revised version of the manuscript that addresses the points raised during the review process.

We look forward to receiving your revised manuscript.

Kind regards,

Yosuke Yamada

Academic Editor

PLOS ONE

Journal Requirements:

"This work was supported by grants from the Research and Development Award Program of the Ministry of Health and Welfare with the award number of PG11001-0566."

"One of the authors (KCH) is employed by Starbia Meditek Co., Ltd. This does not alter our adherence to all the PLoS ONE policies on sharing data and materials. There are no patents, products in development nor marketed products to be declared. The other authors declare no conflict of interest."

We note that one or more of the authors are employed by a commercial company: Starbia Meditek Co., Ltd. 

“The funder provided support in the form of salaries for authors [KCH], but did not have any additional role in the study design, data collection and analysis, decision to publish, or preparation of the manuscript. The specific roles of these authors are articulated in the ‘author contributions’ section.”

(2) Please also provide an updated Competing Interests Statement declaring this commercial affiliation along with any other relevant declarations relating to employment, consultancy, patents, products in development, or marketed products, etc.  

Within your Competing Interests Statement, please confirm that this commercial affiliation does not alter your adherence to all PLOS ONE policies on sharing data and materials by including the following statement: ""This does not alter our adherence to  PLOS ONE policies on sharing data and materials.” (as detailed online in our guide for authors http://journals.plos.org/plosone/s/competing-interests) . If this adherence statement is not accurate and  there are restrictions on sharing of data and/or materials, please state these. Please note that we cannot proceed with consideration of your article until this information has been declared.

6. We note that you have indicated that data from this study are available upon request. PLOS only allows data to be available upon request if there are legal or ethical restrictions on sharing data publicly. For more information on unacceptable data access restrictions, please see http://journals.plos.org/plosone/s/data-availability#loc-unacceptable-data-access-restrictions. 

7. Your ethics statement should only appear in the Methods section of your manuscript. If your ethics statement is written in any section besides the Methods, please move it to the Methods section and delete it from any other section. Please ensure that your ethics statement is included in your manuscript, as the ethics statement entered into the online submission form will not be published alongside your manuscript. 

8. Please include a separate caption for each figure in your manuscript.

9. Please ensure that you refer to Figures 3 and 4 in your text as, if accepted, production will need this reference to link the reader to the figure.

Reviewers' comments:

Reviewer's Responses to Questions

**Comments to the Author**

1. Is the manuscript technically sound, and do the data support the conclusions?

Reviewer #1: Yes

Reviewer #2: No

2. Has the statistical analysis been performed appropriately and rigorously? 

Reviewer #1: Yes

Reviewer #2: Yes

3. Have the authors made all data underlying the findings in their manuscript fully available?

Reviewer #1: Yes

Reviewer #2: Yes

4. Is the manuscript presented in an intelligible fashion and written in standard English?

Reviewer #1: No

Reviewer #2: Yes

5. Review Comments to the Author

Reviewer #1: This is an interesting paper that developed and validate a BIA model for assessing the psoas major muscle in older adults.

I have a few comments:

- English grammar and sentences should be reviewed by a native speaker

- Please be consistent with the use of some abbreviations such as TBI and also bioimpedance (bioelectrical impedance seems more appropriate)

- The abstract is lacking a clear aim/purpose

- Last sentence of the results section in the abstract, change “...and smaller LOA” to “…with small LOA”

- in the fourth paragraph of the introduction, authors should also mention ultrasound. A smooth transition between the differences of using MRI and CT before the sentence about anthropometry is required. A clear and robust methodological reference to the previous literature is needed.

- In the methods section it is missing a sample and power analysis

- I would advise to remove middle-aged adults and to just keep age >60yrs

- At the anthropometry section, please provide a reference for the defined anthropometric measurements

- At the CT method (including reproducibility), it is not clear why the authors include the abdominal visceral and subcutaneous fat areas, as the response variable was PMM

- At the statistical analysis section, please include “respectively” within the sentence beginning as “Two-thirds…”

- At the statistical analysis please consider for the validation analysis the inclusion of i) pure error and ii) test if the slope and intercept did not differ from 1 and 0, respectively (line of identity)

- At the results section, a few abbreviations in tables 1 and 2 need to be defined

- Also, at the results section the authors indicated that height, age, gender, and BIA parameters were included as independent variables but in the statistical section WC, HC, and BMI have also been also mentioned as potential IV. Please revise.

- The discussion needs to be clear and focused. For instances, in the third paragraph the authors mentioned that reproducibility of scan images was performed but the results are not available. In the same paragraph the authors discuss methodological issues about MRI and CT abdominal fat regions which are not relevant for this study

- In the fourth paragraph of the discussion, the inclusion of the r coefficients in the text needs a better sentence structure.

- In the sixth paragraph of the discussion, the BIA method was not used to estimate “the muscles or lean muscles of the trunk” but rather lean soft tissue, as DXA do not provide measurements of muscles or trunk muscles (references 34 and 35)

- In the seventh paragraph the authors mentioned reference 36 to state that multi-frequency BIA provide more reliable results than single frequency for muscle area and volume. Such a statement requires more studies. Also, reference 36 used DXA and therefore, muscle area or volume were not assessed but rather lean soft tissue (or appendicular LST)

- In the last sentence of the seventh paragraph the authors need to clarify what they mean about “a new measurement model must be verified differently to confirm...” Are the authors referring the need to perform an external validation of the PMM BIA equation proposed in this study?

- In the eight paragraph of the discussion, the authors should change “ The present study is..” to “The present equation is..”

- In the same paragraph please clarify the sentence “Therefore, the logic and results…are reasonable”. The authors used BIA and not anthropometry. Please explain and revise

- Again in the same paragraph, please include the word “individual” before “variability” and rephrase the sentence beginning as “Considering…” (the meaning of this sentence is missing)

- Please revise and be focus about the sentences that start from “The acquisition of other … the use of BIA for PMM measurements”. It is confusing, vague, and at some point apparently in opposition to your findings. Please revise accordingly.

- In the last paragraph, the first sentence needs a grammar revision. Also, the authors should include that the findings are only applicable for this single-frequency BIA equipment and that further studies are required to test its applicability to other models/equipment or if a 50 kHz frequency obtained from multi-frequency or BIS equipment were used.

- In this last paragraph of the discussion the author should change the word “race” by “ethnicity” and also change “BIA results” by “BIA equation”

Reviewer #2: The application of BIA has a great advantage in that muscle mass can be easily measured in a clinical setting. Although there are advantages in estimating muscle mass of the whole body and each segment, this approach cannot exactly distinguish individual muscles. The estimation equation of the psoas major muscle using the whole-body impedance includes the effect of other thigh and trunk muscles. I thought that this effect was large for estimating the psoas major muscle CSA. Please explain the theoretical background for predicting the psoas major muscle size from whole-body impedance. Relationships with whole-body or segmental muscle volume and the psoas major muscle CSA should also be mentioned, as psoas major muscle is susceptible to age-related changes（Ikezoe T, et al. Arch Gerontol Geriatr 53, 153-157, 2011）. The estimation equation was created for older people, but did muscle atrophy occur selectively in the psoas major muscle compared to other muscles? In addition, did the psoas major muscle CSA at L3-4 level correlate with its muscle volume? Based on these results, please consider the validity of estimation equation of the psoas major muscle. Furthermore, SEE of the regression equation is converted to %SEE by dividing the SEE values by the mean the psoas major muscle CSA. The %SEE for female and male in the modeling group can be calculated as 19.3% and 12.7%, respectively. Please discuss the accuracy of model fitting and the clinically acceptable measurement error in estimation of the psoas major muscle CSA.

6. PLOS authors have the option to publish the peer review history of their article (what does this mean?). If published, this will include your full peer review and any attached files.

Reviewer #1: No

Reviewer #2: No

---

## [Author Response · Author response to Decision Letter 0]

9 Feb 2023

Response List

Journal Requirements:

Response:

Thank you for your reminder, we have revised the manuscript format as suggested.

Response:

“All included participants provided signed informed consent as required by the Human Experiment Ethics Committee.” of Informed Consent Statement was added in the Methods.

Response:

The source of funding for this study is the Research and Development Award Program of the Ministry of Health and Welfare with the award number of PG11001-0566, which we have disclosed in the Funding information section of the revised manuscript.

"This work was supported by grants from the Research and Development Award Program of the Ministry of Health and Welfare with the award number of PG11001-0566."

Response:

We have added the “The funders had no role in study design, data collection and analysis, decision to publish, or preparation of the manuscript.” in the cover letter。

"One of the authors (KCH) is employed by Starbia Meditek Co., Ltd. This does not alter our adherence to all the PLoS ONE policies on sharing data and materials. There are no patents, products in development nor marketed products to be declared. The other authors declare no conflict of interest."

We note that one or more of the authors are employed by a commercial company: Starbia Meditek Co., Ltd. 

“The funder provided support in the form of salaries for authors [KCH], but did not have any additional role in the study design, data collection and analysis, decision to publish, or preparation of the manuscript. The specific roles of these authors are articulated in the ‘author contributions’ section.”

Response:

We have added the “The Starbia Meditek Co., Ltd provided support in the form of salaries for authors [KCH], but did not have any additional role in the study design, data collection and analysis, decision to publish, or preparation of the manuscript. The specific roles of these authors are articulated in the ‘author contributions’ section.” in the Funding Statement。

(2) Please also provide an updated Competing Interests Statement declaring this commercial affiliation along with any other relevant declarations relating to employment, consultancy, patents, products in development, or marketed products, etc. 

Within your Competing Interests Statement, please confirm that this commercial affiliation does not alter your adherence to all PLOS ONE policies on sharing data and materials by including the following statement: ""This does not alter our adherence to PLOS ONE policies on sharing data and materials.” (as detailed online in our guide for authors http://journals.plos.org/plosone/s/competing-interests) . If this adherence statement is not accurate and there are restrictions on sharing of data and/or materials, please state these. Please note that we cannot proceed with consideration of your article until this information has been declared.

Response:

"This does not alter our adherence to PLOS ONE policies on sharing data and materials." statement has already existed in the Competing Interests Statement in the manuscript.

Response:

“The Starbia Meditek Co., Ltd provided support in the form of salaries for authors [KCH], but did not have any additional role in the study design, data collection and analysis, decision to publish, or preparation of the manuscript. The specific roles of these authors are articulated in the ‘author contributions’ section.” and "This does not alter our adherence to PLOS ONE policies on sharing data and materials." have been added in the cover letter.

6. We note that you have indicated that data from this study are available upon request. PLOS only allows data to be available upon request if there are legal or ethical restrictions on sharing data publicly. For more information on unacceptable data access restrictions, please see http://journals.plos.org/plosone/s/data-availability#loc-unacceptable-data-access-restrictions. 

Response:

“If there are ethical or legal restrictions on sharing a de-identified data set, please explain them in detail (e.g., data contain potentially sensitive information, data are owned by a third-party organization, etc.) and who has imposed them (e.g., an ethics committee). Please also provide contact information for a data access committee, ethics committee, or other institutional body to which data requests may be sent.” have been added in the cover letter.

7. Your ethics statement should only appear in the Methods section of your manuscript. If your ethics statement is written in any section besides the Methods, please move it to the Methods section and delete it from any other section. Please ensure that your ethics statement is included in your manuscript, as the ethics statement entered into the online submission form will not be published alongside your manuscript. 

Response:

The research ethics statement has been moved to the Methods paragraph as revised manuscript.

8. Please include a separate caption for each figure in your manuscript.

Response:

Thank you for your suggestions and reminders, we have added titles to each figure, as indicated in the revised manuscript.

9. Please ensure that you refer to Figures 3 and 4 in your text as, if accepted, production will need this reference to link the reader to the figure.

Response:

Thank you for your reminder, we have added the descriptions of Figure 3 and 4 to the Results Section in the revised manuscript. 

Reviewer #1:

 This is an interesting paper that developed and validate a BIA model for assessing the psoas major muscle in older adults.

I have a few comments:

- English grammar and sentences should be reviewed by a native speaker

Response:

The manuscript has been edited by professionals in English before submission, and the editing proof documents are shown in the attachment. With suggestions from you and other reviewers, the manuscript content and description should have been improved.

- Please be consistent with the use of some abbreviations such as TBI and also bioimpedance (bioelectrical impedance seems more appropriate)

Response:

Thanks for your suggestion. We have checked and corrected abbreviations in the revised manuscript to make them consistent.

- The abstract is lacking a clear aim/purpose

Response:

Thank you for your suggestions and reminders. We have revised the background paragraph in the abstract to "For the assessment of sarcopenia or other geriatric frailty syndromes, psoas area may be one of the primary indicators. Aim to develop and cross-validate the psoas cross-sectional area estimation equation of L3-L4 of the elderly over 60 years old by bioelectrical impedance analysis (BIA)."

- Last sentence of the results section in the abstract, change “...and smaller LOA” to “…with small LOA”

Response:

Thanks for your advice. We have revised the manuscript as suggested.

- in the fourth paragraph of the introduction, authors should also mention ultrasound. A smooth transition between the differences of using MRI and CT before the sentence about anthropometry is required. A clear and robust methodological reference to the previous literature is needed.

Response:

Thank you for your suggestions and reminders. The paragraph has been rewritten as "The use of ultrasound to measure the thickness of the psoas major muscle is a convenient method. However, imaging methods such as computed tomography and magnetic resonance imaging can be reliably used to quantify the size of the psoas major muscle."

- In the methods section it is missing a sample and power analysis

Response:

Thanks for your suggestions. "Power analysis uses the G-Power analysis software program (G Power 3.1.9, University of Dusseldorf, Germany) to determine the minimum number of samples required to establish the most regression equation (calculated with alpha = 0.05, two-tailed, power = 0.95, Number of predictors =4), the minimum sample size is 60 people". The above additions have been added to the "Method section".

- I would advise to remove middle-aged adults and to just keep age >60yrs

Response:

Thanks for your suggestions. We have removed the "middle-aged adults" in the Participants section.

- At the anthropometry section, please provide a reference for the defined anthropometric measurements

Response:

Thanks for your suggestions. We have added references (Lohman et al., 1988) to Anthropometric Methods.

26. Lohman TG, Toche AF, Martoell R. Anthropometric Standardization Reference Manual. Champaign: Human Kinetics, 1988.

- At the CT method (including reproducibility), it is not clear why the authors include the abdominal visceral and subcutaneous fat areas, as the response variable was PMM

Response:

Thank you for your reminder and suggestion, we have deleted the description about “abdominal visceral” and “subcutaneous fat area” in the Reproducibility section, as shown in the revised manuscript.

- At the statistical analysis section, please include “respectively” within the sentence beginning as “Two-thirds…”

Response:

We have corrected as suggested by you, as shown in the revised manuscript.

- At the statistical analysis please consider for the validation analysis the inclusion of i) pure error and ii) test if the slope and intercept did not differ from 1 and 0, respectively (line of identity)

Response:

Thank you for your suggestion and reminder," The pure errors are square root of mean of squares of differences between observed and predicted values. The ordinary least products regression analysis were used to examine the relationship between the PMMCT and PMMBIA and to determine the fixed bias proportional bias [39].” has been added to the statistical analysis section.

"The pure errors obtained by bringing the PMMBIA estimation equation into VG is 2.88 cm2, and the intercept is -4.763 (95% CI is -9.782, 0.254 respectively) and the slope is 1.312 (95% CI is respectively 0.254, 1.632), the curve passes through 0 and 1 respectively.” The statistical results have been added to the Results sections.

27. Ludbrook J. Statistical techniques for comparing measures and methods of measurement: a critical review. Clinical and experimental pharmacology & physiology. 2002;29:527-536.

- At the results section, a few abbreviations in tables 1 and 2 need to be defined

Response:

Thank you for your suggestions and reminders, we have added definition abbreviations in Table 1, 2.

- Also, at the results section the authors indicated that height, age, gender, and BIA parameters were included as independent variables but in the statistical section WC, HC, and BMI have also been also mentioned as potential IV. Please revise.

Response:

Thank you for your suggestions and reminders. We have added variables such as HC, WC and BMI to the description of the Results section.

- The discussion needs to be clear and focused. For instances, in the third paragraph the authors mentioned that reproducibility of scan images was performed but the results are not available. In the same paragraph the authors discuss methodological issues about MRI and CT abdominal fat regions which are not relevant for this study

Response:

Thank you for your suggestion and reminder. We have deleted the third paragraph of the Discussion section.

- In the fourth paragraph of the discussion, the inclusion of the r coefficients in the text needs a better sentence structure.

Response:

Thank you for your suggestions and reminders. We have added the correlation (r) between each predictor variable and PMM to the Results section to make the narrative structure more complete.

- In the sixth paragraph of the discussion, the BIA method was not used to estimate “the muscles or lean muscles of the trunk” but rather lean soft tissue, as DXA do not provide measurements of muscles or trunk muscles (references 34 and 35)

Response:

Thank you for your suggestions and reminders. DXA can only measure soft lean and cannot measure muscle or lean muscles. We have made corrections, as shown in the revised manuscript

- In the seventh paragraph the authors mentioned reference 36 to state that multi-frequency BIA provide more reliable results than single frequency for muscle area and volume. Such a statement requires more studies. Also, reference 36 used DXA and therefore, muscle area or volume were not assessed but rather lean soft tissue (or appendicular LST)

Response:

Thank you for your suggestions and reminders. We have removed "muscle area and volume" from the narrative. Added "Further research is needed to support the findings of this preliminary study" in the Discussion section.

- In the last sentence of the seventh paragraph the authors need to clarify what they mean about “a new measurement model must be verified differently to confirm...” Are the authors referring the need to perform an external validation of the PMM BIA equation proposed in this study?

Response:

Thank you for your suggestions and reminders, we re-expressed the statement as - "The estimation of BIA in PMM was verified by MG in this study, and the correlation and consistency have good results. But this study is Preliminary research on the BIA estimation of PMM. In addition to the consideration of increasing the number of subjects, it is also necessary to confirm the repeatability of this research through research in other laboratories, so as to increase the application value of BIA in PMM."

- In the eight paragraph of the discussion, the authors should change “ The present study is..” to “The present equation is..”

Response: 

Thank you for your reminders and suggestions, we corrected as indicated in the revised manuscript.

- In the same paragraph please clarify the sentence “Therefore, the logic and results…are reasonable”. The authors used BIA and not anthropometry. Please explain and revise

Response: 

Thank you for your suggestion and reminder, we corrected the anthropometry in this narrative to BIA, as indicated in the revised manuscript.

- Again in the same paragraph, please include the word “individual” before “variability” and rephrase the sentence beginning as “Considering…” (the meaning of this sentence is missing)

Response: 

Thank you for the reminder, and your kind suggestions are reflected in the revised manuscript.

- Please revise and be focus about the sentences that start from “The acquisition of other … the use of BIA for PMM measurements”. It is confusing, vague, and at some point apparently in opposition to your findings. Please revise accordingly.

Response: 

Thank you for your suggestions and reminders, we have rewritten this paragraph as "This study still has limitations on the measurement accuracy of PMM, but it is expected that adding other measurement parameters to the existing single-frequency BIA estimation equation may increase its measurement accuracy. In the future, it can be considered to add measurement parameters such as body circumference measurement in 3D human body scanning [43, 44], local impedance measurement of the waist or back [45], or multi-frequency impedance or bio-impedance spectrum analysis [36] to optimize the measurement model of PMM."

- In the last paragraph, the first sentence needs a grammar revision. Also, the authors should include that the findings are only applicable for this single-frequency BIA equipment and that further studies are required to test its applicability to other models/equipment or if a 50 kHz frequency obtained from multi-frequency or BIS equipment were used.

Response: 

Thank you for the reminder, and your kind suggestions are reflected in the revised manuscript.

- In this last paragraph of the discussion the author should change the word “race” by “ethnicity” and also change “BIA results” by “BIA equation”

Response:

Thank you for the reminder, and your kind suggestions are reflected in the revised manuscript. 

Reviewer #2:

 The application of BIA has a great advantage in that muscle mass can be easily measured in a clinical setting. Although there are advantages in estimating muscle mass of the whole body and each segment, this approach cannot exactly distinguish individual muscles. 

Response: 

Thank you for your suggestions. We have added this narrative to the Discussion section to make this research more complete.

The estimation equation of the psoas major muscle using the whole-body impedance includes the effect of other thigh and trunk muscles. I thought that this effect was large for estimating the psoas major muscle CSA.

Response: 

Thank you for your suggestions and comments. "Using WBI, are the variables of thigh and trunk muscles related to the estimation of psoas major muscle mass? In the future, data obtained from tools such as DXA, CT, and multi-limb segment BIA can be used to conduct correlation research." Added this paragraph to the Discussion section.

Please explain the theoretical background for predicting the psoas major muscle size from whole-body impedance. Relationships with whole-body or segmental muscle volume and the psoas major muscle CSA should also be mentioned, as psoas major muscle is susceptible to age-related changes（Ikezoe T, et al. Arch Gerontol Geriatr 53, 153-157, 2011.

Response:

Thank you for your suggestions and reminders. It has been confirmed that WBI can be used to estimate the lean mass of the whole body [47]. The lean mass of the whole body has a certain correlation with the psoas major [45]. Generally speaking, the muscle mass of the lower limbs will significantly decrease with increasing age. Ikezoe et al. [48] measured the thickness of 10 muscles of the lower extremity, showing minimal atrophy in the soleus muscle. Soleus muscle thickness did not differ significantly between young and old. Among the lower extremity skeletal muscles, the psoas has the highest rate of loss with age. " Added this narrative to the Discussion section.

47. Segal KR, Loan MV, Fitzgerald PI, Hodgdon JA, Itallie TB. Lean body mass estimation by bioelectrical impedance analysis: a four-site cross-validation study. American Journal of Clinical Nutrition. 1988; 478: 7-14. 

48. Ikezoe T, Natsuko M, Nakamura M, Ichihashi N. Age-related muscle atrophy in the lower extremities and daily physical activity in elderly. Archives of Gerontology and Geriatrics. 2011; 53: e153-e157. 

The estimation equation was created for older people, but did muscle atrophy occur selectively in the psoas major muscle compared to other muscles? In addition, did the psoas major muscle CSA at L3-4 level correlate with its muscle volume? 

Response:

Thank you for your reminder and suggestion. "Compared with hospitalized elderly injured patients, the cross-sectional area of Psoas major muscle increases by 1 cm2, and the dependent living decreases by 20% [44], which is of great importance in its application. Psoas major muscle CSA in L3-L4 Level relative to other muscles, such as Psoas major muscle in L4-L5, L5-S1, Multifidus in L3-L4, L4-L5 and L5-S1, and the correlation of skeletal muscle are 0.877, 0.646, 0.500, 0.585, 0.596, and 0.723, respectively [45]". Add this narrative to the discussion section.

44. Fairchild B, Webb TP, Xiang Q, Tarima S, Brasel. Sarcopenia and frailty in elderly trauma patients. World J Surg. 2015;39:373-379.

45. Hiyama A, Katoh H, Sakai D, Tanka M, Sato M, Watanabe M. The correlation analysis between sagittal alignment and cross-sectional area of paraspinal muscle in patient with lmbar spinal stenosis and degenerative spondylolisthesis. BMC Musculoskeletal Disorders. 2019;20:352.

Based on these results, please consider the validity of estimation equation of the psoas major muscle. Furthermore, SEE of the regression equation is converted to %SEE by dividing the SEE values by the mean the psoas major muscle CSA. The %SEE for female and male in the modeling group can be calculated as 19.3% and 12.7%, respectively. Please discuss the accuracy of model fitting and the clinically acceptable measurement error in estimation of the psoas major muscle CSA.

Response:

Thank you for your suggestions and reminders. "Based on studies of quantitative indicators of generalized muscle atrophy in patient populations, psoas measurement has not yet to pass the test of all technical and clinical value. Of the numerous studies on psoas sarcopenia, it has been established that the need for standardized thresholds are important representations for predicting morbidity or mortality, but currently these thresholds are reported in different units of measurement. Therefore, general conclusions cannot be drawn [46]. According to the estimation equation of Psoas major muscle in this study, the SEE% of women and men are 19.3% and 12.7%, respectively. This result still has a certain space for practical clinical application". This paragraph has been added to the Discussion section.

46. Baracos W. Psoas as a sentinel muscle for sarcopenia: a flawed premise. Journal of Cachexia, Sarcopenia and muscle. 2017; 8:527-528.

---

## [Decision Letter · Decision Letter 1]

13 Mar 2023

Preliminary study on the application of bioimpedance analysis to measure the psoas major muscle in older adults

PONE-D-22-26490R1

Dear Dr. Hsieh,

We’re pleased to inform you that your manuscript has been judged scientifically suitable for publication and will be formally accepted for publication once it meets all outstanding technical requirements.

Kind regards,

Yosuke Yamada

Academic Editor

PLOS ONE

Additional Editor Comments (optional):

Reviewers' comments:

Reviewer's Responses to Questions

**Comments to the Author**

1. If the authors have adequately addressed your comments raised in a previous round of review and you feel that this manuscript is now acceptable for publication, you may indicate that here to bypass the “Comments to the Author” section, enter your conflict of interest statement in the “Confidential to Editor” section, and submit your "Accept" recommendation.

Reviewer #1: All comments have been addressed

Reviewer #2: All comments have been addressed

2. Is the manuscript technically sound, and do the data support the conclusions?

Reviewer #1: Yes

Reviewer #2: Yes

3. Has the statistical analysis been performed appropriately and rigorously? 

Reviewer #1: Yes

Reviewer #2: Yes

4. Have the authors made all data underlying the findings in their manuscript fully available?

Reviewer #1: Yes

Reviewer #2: Yes

5. Is the manuscript presented in an intelligible fashion and written in standard English?

Reviewer #1: Yes

Reviewer #2: Yes

6. Review Comments to the Author

Reviewer #1: The authors addressed all the comments and the paper is now improved. I have no additional questions

Reviewer #2: I think that this manuscript has been appropriately revised according to the reviewer's suggestions.

7. PLOS authors have the option to publish the peer review history of their article (what does this mean?). If published, this will include your full peer review and any attached files.

Reviewer #1: No

Reviewer #2: **Yes: **Masashi Taniguchi

---

## [Editor Report · Acceptance letter]

20 Mar 2023

PONE-D-22-26490R1 

Preliminary study on the application of bioimpedance analysis to measure the psoas major muscle in older adults 

Dear Dr. Hsieh:

I'm pleased to inform you that your manuscript has been deemed suitable for publication in PLOS ONE. Congratulations! Your manuscript is now with our production department. 

Kind regards, 

on behalf of

Dr. Yosuke Yamada 

Academic Editor

PLOS ONE